



# Measurement report: Airborne measurements of NO$_x$ fluxes over Los Angeles during the RECAP-CA 2021 campaign

Clara M. Nussbaumer[1,2], Bryan K. Place[2], Qindan Zhu[3], Eva Y. Pfannerstill[4], Paul Wooldridge[2], Benjamin C. Schulze[5], Caleb Arata[4], Ryan Ward[5], Anthony Bucholtz[6], John H. Seinfeld[5], Allen H. Goldstein[4,7], and Ronald C. Cohen[2,3]

[1]Department of Atmospheric Chemistry, Max Planck Institute for Chemistry, 55128 Mainz, Germany
[2]Department of Chemistry, University of California, Berkeley, Berkeley, CA 94720, USA
[3]Department of Earth and Planetary Science, University of California, Berkeley, Berkeley, CA 94720, USA
[4]Department of Environmental Science, Policy, and Management, University of California, Berkeley, Berkeley, CA 94720, USA
[5]Department of Environmental Science and Engineering, California Institute of Technology, Pasadena, CA 91125, USA
[6]Department of Meteorology, Naval Postgraduate School, Monterey, CA 93943, USA
[7]Department of Civil and Environmental Engineering, University of California, Berkeley, Berkeley, CA 94720, USA

**Correspondence:** Clara M. Nussbaumer (clara.nussbaumer@mpic.de) and Ron C. Cohen (rccohen@berkeley.edu)

**Abstract.**

Nitrogen oxides (NO$_x$ $\equiv$ NO + NO$_2$) are involved in most atmospheric photochemistry, including the formation of tropospheric ozone (O$_3$). While various methods exist to accurately measure NO$_x$ concentrations, it is still a challenge to quantify the source and flux of NO$_x$ emissions. We present airborne measurements of NO$_x$ and winds used to infer the emission of 5 NO$_x$ across Los Angeles. The measurements were obtained during the research aircraft campaign RECAP-CA (Re-Evaluating the Chemistry of Air Pollutants in CAlifornia) in June 2021. Geographic allocations of the fluxes are compared to the NO$_x$ emission inventory from the California Air Resources Board (CARB). We find that the NO$_x$ fluxes have a pronounced weekend effect and are highest in the Eastern part of the San Bernardino valley. The comparison of the RECAP-CA and the modeled CARB NO$_x$ fluxes suggest the modeled emissions are too high near the coast and in downtown Los Angeles and too low 10 further inland in the Eastern part of the San Bernardino valley.

## 1 Introduction

Nitrogen oxides (NO$_x$), representing the sum of nitric oxide (NO) and nitrogen dioxide (NO$_2$) are hazardous pollutants and precursor to tropospheric ozone, which is known to have adverse health effects on humans and plants (Boningari and Smirniotis, 2016; Mills et al., 2018; Nuvolone et al., 2018; CARB, 2022b). NO$_x$ is emitted from some natural sources including soil, 15 microbial activity and lightning, but mostly from anthropogenic combustion sources, such as electricity generation facilities and motor vehicles, with the latter dominating in urban environments (Delmas et al., 1997; Pusede et al., 2015). Densely populated cities, such as the megacity Los Angeles, often suffer from poor air quality leading to increases in respiratory diseases and premature mortality (Stewart et al., 2017). Air quality monitoring, public policy and new emission control technologies have



been developed and implemented to assess, guide and manage emissions, leading to healthier air in cities (CARB, 2022b).
Significant reductions in $NO_x$ and other primary pollutants have occurred in the U.S. and specifically in Los Angeles over the past decades (e.g. Qian et al. (2019); Nussbaumer and Cohen (2020); EPA (2022a)). However, ozone exceedances of the national ambient air quality standard (NAAQS) of 0.070 ppm (8-hr maximum) are still frequent in Los Angeles which the American Lung Association (2022) found to have the highest ozone pollution in all of the United States (South Coast Air Quality Management District, 2017; EPA, 2022c). In the summer months June - September of 2021, $O_3$ exceeded the NAAQS on more than half the days. Exceedances were even more frequent in 2020, demonstrating that further precursor reductions are imperative (EPA, 2022b).

The combination of emission inventories with models provide insight into the emission reductions needed to achieve healthy air and synthesize our understanding of atmospheric dynamics, that describe the flow of emissions from their source through the atmosphere, the deposition of chemicals to the earth's surface and the oxidation of molecules. Comparison of predicted concentrations of chemicals with this type of combined modeling system provides a guide for the needed reductions in emissions to protect public health. Errors and biases in any part of this system can lead to incorrect estimates of the amount of reduction needed to achieve a particular goal. Fujita et al. (2013) pointed towards the problem of biased emission inventories regarding future predictions of ozone, naming underestimations in early VOC (volatile organic compound) SoCAB (South Coast Air Basin) emission inventories and hence modeled VOC/$NO_x$ that did not match the atmosphere as a key bias in understanding ozone chemistry.

Direct observational mapping of emissions would allow a more straightforward evaluation of the accuracy of emission inventories without requiring untangling potential errors in emissions from those of transport or chemistry. Until recently, such measurements have been rare because of the difficulty of obtaining and interpreting measurements of emissions in heterogeneous urban environments and especially the difficulty of mapping emissions over the spatial scales needed to assess the full complement of urban processes. Recently, several experiments have overcome these challenges, measuring $NO_x$ and VOC fluxes using aircraft platforms.

Airborne studies on VOC fluxes include Karl et al. (2013) and Misztal et al. (2014) who present fluxes of biogenic VOCs based on research flights during the CABERNET (California Airborne BVOC Emission Research in Natural Ecosystem Transects) campaign over Californian oak forests in June 2011. Yuan et al. (2015) determined $CH_4$ and VOC emissions from two shale gas production plants in the Southern United States based on aircraft measurements in summer 2013. Yu et al. (2017) derived isoprene and monoterpene fluxes during the airborne Southeast Atmosphere Study above the U.S. in 2013. Other studies presenting VOC fluxes based on aircraft measurements include Karl et al. (2009), Conley et al. (2009), Kaser et al. (2015), Wolfe et al. (2015),Gu et al. (2017) and Yu et al. (2017).

Nitrogen oxides fluxes based on airborne measurements are even less common than VOCs. Wolfe et al. (2015) reported $NO_x$ fluxes based on a measurement flight during the NASA SEAC[4]RS campaign in 2013. Vaughan et al. (2016) and Vaughan et al. (2021) reported $NO_x$ fluxes based on this method over the Greater London region during the OPFUE (Ozone Precursors Fluxes in an Urban Environment) campaign in July 2014. They compared emission predictions from the National Atmospheric Emissions Inventory with the calculated $NO_x$ fluxes via wavelet transformation, which they found to be higher than the inventory



by up to a factor of 2, underlining the importance of emission inventory validation. Zhu et al. (2023) recently reported $NO_x$
fluxes over the San Joaquin Valley of California based on the RECAP-CA (Re-Evaluating the Chemistry of Air Pollutants in
CAlifornia) aircraft campaign where $NO_x$ from soils was identified key contributor to the overall emission.

In this paper, we present $NO_x$ flux calculations via wavelet transformation from aircraft measurements of $NO_x$ concentrations and the vertical wind speed during the RECAP-CA aircraft campaign over Los Angeles in June 2021. We provide
footprint calculations for investigating the origin of the sampled air masses and compare our results to the emission inventory
of the California Air Resources Board (CARB).

## 2 Observations and methods

### 2.1 RECAP-CA aircraft campaign

The RECAP-CA (Re-Evaluating the Chemistry of Air Pollutants in CAlifornia) aircraft campaign took place in June 2021
over Los Angeles and the Central Valley of California with the campaign base in Burbank, California (34.20 $^\circ$ N, 118.36 $^\circ$ W)
using the CIRPAS (Center for Interdisciplinary Remotely Piloted Aircraft Studies) Twin Otter aircraft. Details on the research
aircraft (Figure S1) can be found in Reid et al. (2001) and Hegg et al. (2005). The ambient air was sampled with an inlet
approx. 1 m above the aircraft nose at a sampling speed of around 60 m s$^{-1}$. The aircraft carried instruments for measurements
of meteorological data, nitrogen oxides, volatile organic compounds and greenhouse gases. All flights were carried out between
11:00 and 18:00 local time (LT).

We focus on the measurements over Los Angeles which took place on three weekends (June 6, 12 and 19) and six weekdays
(June 1, 4, 10, 11, 18 and 21) in 2021. The flight paths are shown in Figure 1a. All flights were carried out at an altitude of
roughly 300 - 400 m above ground level, covering the coastal region of Los Angeles, parts of Santa Ana county, Los Angeles
Downtown and the San Bernardino valley. Flight days were chosen to explore as wide a range of temperature as possible. In
addition, about half the flights started with the East-West legs and the other half the coastal North-South legs to gain additional
variation in temperature. For further analysis, we separated the covered area in four different segments, as shown in Figure 1b.

### 2.2 Meteorological measurements

The meteorological instruments on-board the CIRPAS Twin Otter research aircraft were previously described in Karl et al.
(2013). Temperature was obtained by a Rosemount Sensor (Emerson Electric Co., St. Louis, Missouri, USA). Dew point
temperature was measured by a DewMaster Chilled Mirror Hygrometer (Edgetech Instruments Inc., Hudson, Massachusetts,
USA). Differential and barometric pressure sensors (Setra Systems Inc., Boxborough, Massachusetts, USA) were used to
determine the pressure. GPS latitude, longitude, altitude, ground speed and track, as well as pitch, heading and roll angle were
measured by a C-MIGITS (Miniature Integrated GPS (Global Positioning System)/INS (Inertial Navigation System) Tactical
System) III (Systron Donner Inertial Division, Concord, California, USA). A radome flow angle probe provided the true air
speed and the 3D wind.





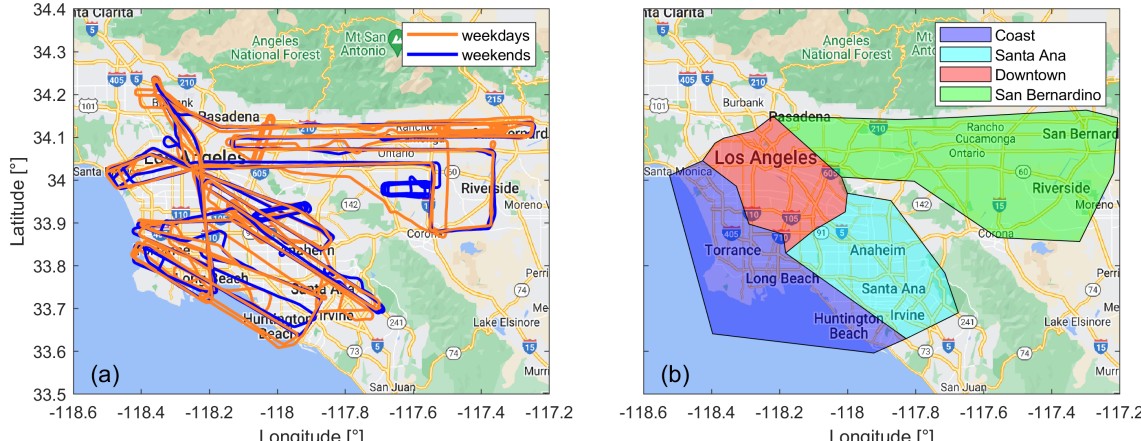

**Figure 1.** (a) Overview of the flight paths during the RECAP-CA campaign over Los Angeles in June 2021. Blue colors show weekend flights and orange colors show weekday flights. (b) Geographic separation of the covered area in four segments, including the coast (blue), parts of Santa Ana county (light blue), Downtown Los Angeles (red) and the San Bernardino valley (green). © Google Maps 2023.

## 2.3 $NO_x$ measurements

$NO_x$ measurements were carried out using a custom-built three-channel thermal dissociation laser-induced fluorescence (TD-LIF) instrument with a detection limit of $\sim 15$ pptv (10 s) and a precision ($2\,\sigma$) of $< 7\,\%$ (Thornton et al., 2000). The instrument is described in detail in Thornton et al. (2000), Day et al. (2002) and Sparks et al. (2019). Briefly, $NO_2$ is excited in the first channel with a 532 nm $Nd^{3+}$:$YVO_4$ laser (Explorer One XP, Spectra Physics). The fluorescence resulting from $NO_2^*$ de-excitation is detected by a photomultiplier tube as a signal which is approximately proportional to the ambient $NO_2$ mixing ratio. The proportionality arises because the fluorescence signal and the quenching of the fluorescence both scale with pressure. Calibration with an $NO_2$ gas standard (5.5 ppm; Praxair) was performed once an hour. The instrument background was determined every 20 minutes using scrubbed ambient air. NO is determined in the second channel through conversion to $NO_2$ by adding excess ozone ($O_3$). $NO_y$ species were detected through thermal dissociation at $\sim 500\,^\circ$C to $NO_2$ in the third channel (Day et al., 2002). Ambient air was sampled at a flow rate of 6 l/min, equally divided into the three instrument channels.

## 2.4 $NO_x$ flux calculations

The emission of a trace gas is characterized as a flux which is a mass emitted per area and time, e.g. mg m$^2$ h$^{-1}$. A flux $F$ can be described as the covariance of the fluctuation in the vertical wind speed $\omega'$ and the fluctuation in the concentration of the trace gas of interest $c'$, as shown in Eq. (1).

$$F = \overline{\omega' c'} \tag{1}$$





Analyses of the covariance of winds and concentrations from tower-based or aircraft-based observations enable the determination of fluxes. With the eddy covariance method, the flux is directly calculated from the measurements as the mean of the product of the deviation of the vertical windspeed from the mean of the vertical wind speed and the deviation of the concentration analogously, as shown in Eq. (2) (e.g. Schaller et al. (2017); Desjardins et al. (2021)).

$$\overline{\omega'c'} = \frac{1}{N}\sum_{x=1}^{N}(\omega_x - \overline{\omega})(c_x - \overline{c}) \tag{2}$$

Requirements for accurate fluxes with EC are steady state conditions and homogeneous horizontal air masses. Typically an averaging time of at least 30 minutes is used to ensure the full spectrum of eddies are sampled (Schaller et al., 2017; Desjardins et al., 2021). The 30 minutes long averaging time is easily implementable for stationary tower installments. For aircraft observations, the high aircraft velocity and the associated rapid geographical change are inconsistent with the steady state requirement. An alternative is the flux calculation via wavelet transformation. This approach does not require the assumption of stationary conditions as it enables the determination of a flux localized both in time and frequency (Karl et al., 2013; Schaller et al., 2017). A discrete time series such as the vertical wind speed or the concentration of an atmospheric trace gas is convolved with a wavelet function $\psi_{a,b}(t)$ yielding the wavelet coefficient $W(a,b)$ according to Eq. (3) (Torrence and Compo, 1998; Thomas and Foken, 2005; Schaller et al., 2017; Desjardins et al., 2021).

$$W(a,b) = \int_{-\infty}^{\infty} x(t)\psi_{a,b}(t)dt \tag{3}$$

The function $\psi_{a,b}(t)$, Eq. (4), a wavelet, scaled by $a$ and shifted by $b$, controlling the frequency and the time of the wavelet, respectively. In our study, we use the complex Morlet wavelet $\psi_M$, Eq. (5), which is the product of a sine and a gaussian function, with $\omega_0 = 6$ and $u = \frac{t-b}{a}$ (Torrence and Compo, 1998; Metzger et al., 2013; Wolfe et al., 2018).

$$\psi_{a,b}(t) = \frac{1}{\sqrt{a}} \times \psi_M(\frac{t-b}{a}) \tag{4}$$

$$\psi_M(u) = \pi^{-\frac{1}{4}} \times e^{-i\omega_0 u} \times e^{-\frac{u^2}{2}} \tag{5}$$

The expression $|W^2(a,b)|$ represents the power spectrum of the discrete time series, showing each scale $a$ and translation $b$ with the according amplitude. For two different time series, the product of the wavelet coefficients yields the cross power spectrum which when integrated represents the covariance and, in case of the vertical wind speed and the trace gas concentration, the flux (Torrence and Compo, 1998; Metzger et al., 2013; Schaller et al., 2017). The flux is calculated through wavelet

transformation via Eq. (6) whereas $C_\delta$ is a reconstruction factor equal to 0.776 for the Morlet wavelet. $N$ is the number of



elements in the time series ($n = 0, 1, ..., N$) with the time step $\delta t$ and $J$ is the number of scales ($j = 0, 1, ..., J$) with the spacing $\delta j$ (Metzger et al., 2013; Schaller et al., 2017; Desjardins et al., 2021).

$$\overline{\omega'c'} = \frac{\delta t}{C_\delta} \times \frac{\delta j}{N} \times \sum_{n=0}^{N-1} \sum_{j=0}^{J} \frac{[W_c(a,b) \times W_\omega^*(a,b)]}{a(j)} \tag{6}$$

Both data pre-treatment (of the $NO_x$ and meteorological data) and wavelet analysis were performed following the procedure
presented by Vaughan et al. (2021). The time stamp of the meteorological data was interpolated to the $NO_x$ data time stamp with a resolution of 5 Hz. We generated flight segments as input for the wavelet analysis with a 10 km minimum length of continuous measurements. Observations above the boundary layer and thus in the free troposphere were excluded from our analysis as they are assumed to be out of contact with the surface below. Data with an aircraft roll angle larger than $8°$ or where the altitude changed by more than $\sim 100$ m across the 10 km were excluded. Due to different inlet sampling locations
and computer clocks, the $NO_x$ measurements and the vertical wind speed measurements were slightly time shifted. This lag correction was quantified via the cross covariance of the two time series. The vertical wind speed was then shifted to the $NO_x$ measurements according to the covariance peak. We used the median lag of all segments of the same flight day for the lag correction. In Figure S2 of the Supplement we show an example covariance peak for $NO_x$ and potential temperature $\theta$ with the vertical wind speed, respectively, for three segments on June 6.

For the wavelet analysis, we followed the procedure described in detail in Torrence and Compo (1998). The wavelet transform was calculated separately for the vertical wind speed fluctuation $\omega'$ and the $NO_x$ concentration fluctuation $c'$. We used the wavelet software provided by C. Torrence and G. Compo, with the Morlet wavelet $\psi_M$ (described in Eq. (5)), the time step $\delta t = 0.2\,\mathrm{s}^{-1}$, a scale spacing $\delta j = 0.25$ and a scale number of $J = log_2(\frac{N\delta t}{j_{min}}) \times \frac{1}{\delta j}$ (default value, $j_{min} = 2 \times \delta t$). For example, there would be 36 scales for a segment with 1000 data points. The cross spectrum was obtained through the sum of
the product of the real parts of the wavelet transform for $c'$ and $\omega'$ and the product of the imaginary parts, which gave the $NO_x$ flux via the weighted sum over all scales according to Eq. (24) in Torrence and Compo (1998). Due to edge effects, the error is particularly high at the beginning and end of each segment which is described by the cone of influence (COI) (Torrence and Compo, 1998). We have discarded data points for which more than 80 % of the spectral information is located within the COI. Note that approximately 50 % of the data points are lost due to edge effects, likely due to short segment lengths and frequent
calibrations. For our analysis, we used the 2 km moving mean of the $NO_x$ flux. Figure 2a shows the discrete time series of the $NO_x$ concentration and the vertical wind speed for one segment on 6 June with a length of 30.0 km. The x axis shows the time in seconds from the start of the segment ($\sim 9$ min). Figure 2b shows the cross power spectrum of $c'$ and $\omega'$, with red colors representing positive and blue colors representing negative amplitudes, and the cone of influence by the black dashed line. The 2 km moving mean of the resulting $NO_x$ flux is presented in Figure 2c. We show an example co-spectrum for the $NO_x$ flux and
the heat flux in Figure S3 of the Supplement, for three segments on June 6 corresponding to Figure S2.

The overall uncertainty of the calculated $NO_x$ flux is composed of the uncertainty of the measurement of the $NO_x$ concentration and the vertical wind speed as well as the uncertainty associated with the presented method of performing the wavelet transformation, including random and systematic errors (Lenschow et al., 1994; Mann and Lenschow, 1994; Wolfe et al., 2015;





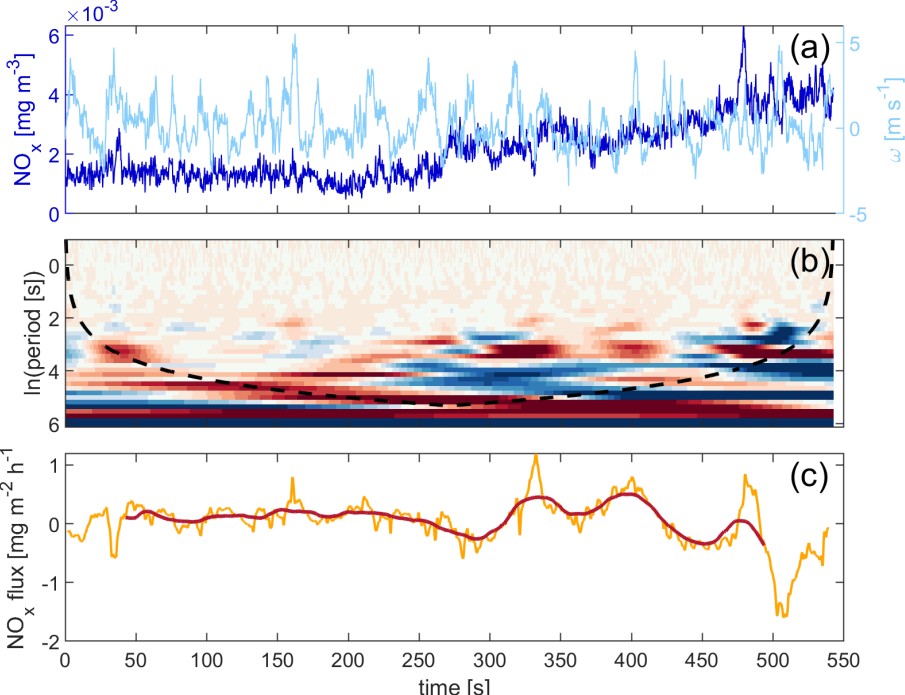

**Figure 2.** (a) Discrete time series of the $NO_x$ concentration (dark blue) and the vertical wind speed (light blue) for a segment on 6 June. (b) Cross power spectrum with 41 scales and 2777 translations, the latter equaling the number of points in the segment. Red colors represent positive and blue colors represent negative amplitudes. (c) The resulting $NO_x$ flux (orange) and the 2 km moving mean (red).

Vaughan et al., 2021). A detailed error analysis for these observations is provided in Zhu et al. (2023). We find the median
and average values of the $NO_x$ flux are dominated by the atmospheric variability and not the measurement uncertainties. The atmospheric variability of $NO_x$ is in the order of 30 % ($1\,\sigma$) which is around 4 times higher compared to the instrumental precision of < 7 % ($1\,\sigma$).

## 2.5    Vertical Divergence

Vertical flux divergence describes the effect that a flux measured at a certain altitude can differ from the surface flux caused for
example through chemistry conversions, entrainment from above or horizontal advection, mostly through differing wind speeds with altitude (Wolfe et al., 2018; Vaughan et al., 2021). The characterization of the vertical divergence can be performed by measurements of a vertical profile over a homogeneous surface. Several race tracks stacked at multiple heights were conducted over Los Angeles during RECAP-CA, but none fulfilled the criteria for performing flux calculations as described in Section 2.4 (e.g. roll angle or segment length). A different approach is described in Wolfe et al. (2018) and Zhu et al. (2023) which investi-
gates the calculated $NO_x$ fluxes in dependence of the relative measurement position in the boundary layer over a homogeneous surface. In contrast to the San Joaquin Valley considered in Zhu et al. (2023), the measurements over Los Angeles are not





homogeneous, neither in space nor in time, due to a high variety of emission sources and a diurnal cycle affected e.g. by rush
hour traffic. In Figure S4 of the Supplement we show the calculated $NO_x$ flux across Los Angeles versus the dimensionless
altitude $z/z_i$, where $z$ is the radar altitude of the research aircraft and $z_i$ the boundary layer height. The data points exhibit a
decreasing trend (green) with altitude pointing towards the effect of vertical divergence, but show a low statistical significance
with an $R^2$ of only 6 %, likely due to the source heterogeneity, both space- and time-wise. In order to investigate the influence
of vertical divergence, we exemplary perform a correction of the fluxes with the fit derived from Figure S4, using a correction
factor as presented in Eq. (15) in Wolfe et al. (2018). The resulting surface flux $F_0$ can then be calculated as shown in Equation
(7), with the measured flux $F_z$ at the altitude $z/z_i$, the slope $m$ of the linear fit and its y-intercept $c$.

$$F_0 = \frac{F_z}{1 + \frac{m}{c} * z/z_i} \tag{7}$$

We show the fluxes adjusted for this estimate of the vertical divergence versus the dimensionless altitude in Figure S5 of the
Supplement. Data points which are located close to the linear fit can be corrected quite accurately. However, corrections for
data points which are located further away from the fit, and particularly those measured close to the boundary layer height, are
highly uncertain. As the slope is negative, and the absolute values for slope and y-intercept are almost equal, the denominator
in Equation (7) gets extremely small close to the BLH and the correction correspondingly large. Karl et al. (2013) also suggests
that fluxes can get uncertain close to the boundary layer height due to entrainment to the free troposphere. Thus for the
sensitivity study, we omit data points within the upper 20 % of the boundary layer ($z/z_i \geq 0.8$).

## 2.6 Footprint calculations

In order to map emissions, we performed footprint calculations which help to identify the areas where the measured air masses
originated (Vesala et al., 2008). We used the footprint model KL04-2D, proposed by Kljun et al. (2004), and further devel-
oped by Metzger et al. (2012) to include the impact of cross winds. The KL04 model was developed from the 3D backward
Lagrangian model KL02 (Kljun et al., 2002). This model was previously applied by Vaughan et al. (2021).

We subdivided the area of observations into a 500 m × 500 m spatial grid and calculated an average of all data points located
in one grid box, separately for each segment. We then performed the footprint calculation for each grid box of each segment,
approximately 5,000 footprints for the entire campaign. The footprint calculation is dependent on the wind direction [°], the
crosswind fluctuations (standard deviation of the horizontal wind speed) [m/s], the vertical wind fluctuations (standard devi-
ation of the vertical wind speed) [m/s], the friction velocity [m/s], the roughness length [m], the altitude of the measurement
above ground level [m] and the height of the planetary boundary layer [m]. The roughness length $z_0$ is a measure of the surface
properties, which we adapted from Burian et al. (2002) and from the World Meteorological Organization (2018) based on
the land cover use for Los Angeles. We first generated footprints using the roughness length from the High-Resolution Rapid
Refresh (HRRR) model by the National Oceanic and Atmospheric Administration (NOAA, 2021). Based on the dominant land
cover type in each footprint, we then applied the roughness length according to Burian et al. (2002) and the World Meteoro-
logical Organization (2018). The land use data set was obtained from the Multi-Resolution Land Characteristics Consortium



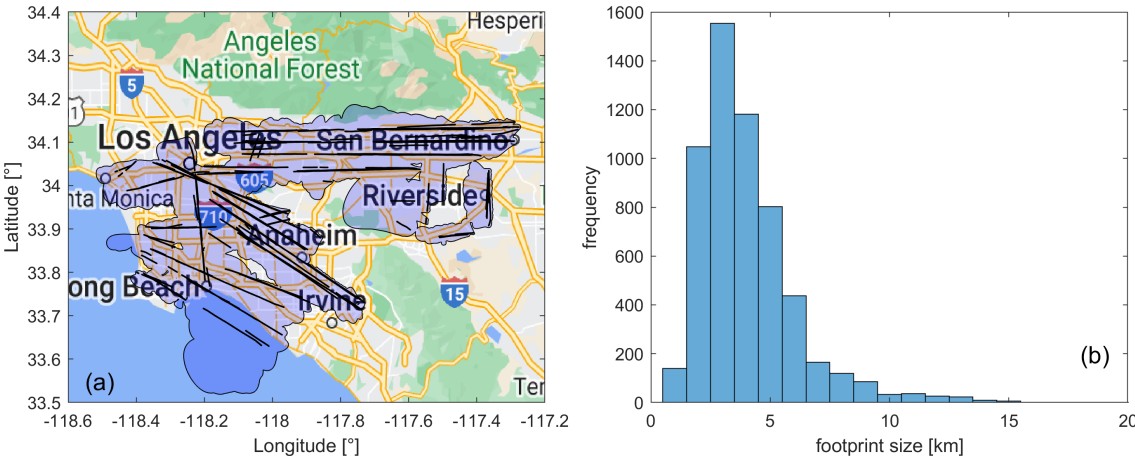

**Figure 3.** (a) Overview of the 90 % footprint influences for the flight campaign over Los Angeles. © Google Maps 2023. (b) Frequency distribution of the 90 % footprint size which describes the distance perpendicular to the flight track.

(MRLC, 2019). The friction velocity $u^*$ is a measure for the shear stress and can be approximated via the logarithmic wind
profile as shown in Eq. (8), where $u$ is the horizontal wind speed, $k$ the Karman constant equaling 0.41 and $z$ the altitude above ground level (Weber, 1999).

$$u^* = u \times k \times ln(\frac{z}{z_0})^{-1} \qquad (8)$$

The model output is a spatial grid of the fractional contribution to each footprint normalized to a value of unity. We focused on the 90 % footprint influences and assigned the measured $NO_x$ flux to the $NO_x$ flux of each grid box in the 90 % footprints.
We then overlayed all footprint grids separately for weekends and for weekdays and calculated a value for the $NO_x$ flux for each grid box as the weighted average.

Figure 3a presents the 90 % contours of all segments over Los Angeles in blue and the according flight paths in black. The majority of the 90 % footprints captured air masses from a distance of $\sim 3$ km (perpendicular to the flight track) as shown in the histogram of footprint size in Figure 3b. We observed individual footprints with a size of up to 23 km, for example for 11 June
around Redondo Beach which were accompanied by high horizontal wind speeds ($\sim 4\,\mathrm{m\,s^{-1}}$) and a flight altitude ($\sim 440$ m) in the upper part of the boundary layer (BLH of $\sim 545$ m). This is in line with the findings by Kljun et al. (2004), demonstrating the impact of the receptor height (Fig. 1 in Kljun et al. (2004)). The smallest footprint with 500 m was observed on 19 June, characterized by a small value for the horizontal wind speed with $\sim 0.5\,\mathrm{m\,s^{-1}}$.

The influence of the horizontal wind speed on the footprint analysis is also highlighted in Figure 4. The two panels present
selected flight segments in geographic proximity over the San Bernardino valley on 6 June (Figure 4a) and on 12 June (Figure 4b). Both days were weekend days and we expect similar $NO_x$ emissions. However, the calculated $NO_x$ flux for the displayed segments was on average $0.18 \pm 0.25\,\mathrm{mg\,m^{-2}\,h^{-1}}$ for 6 June and $1.66 \pm 1.06\,\mathrm{mg\,m^{-2}\,h^{-1}}$ for 12 June. At the same time, the



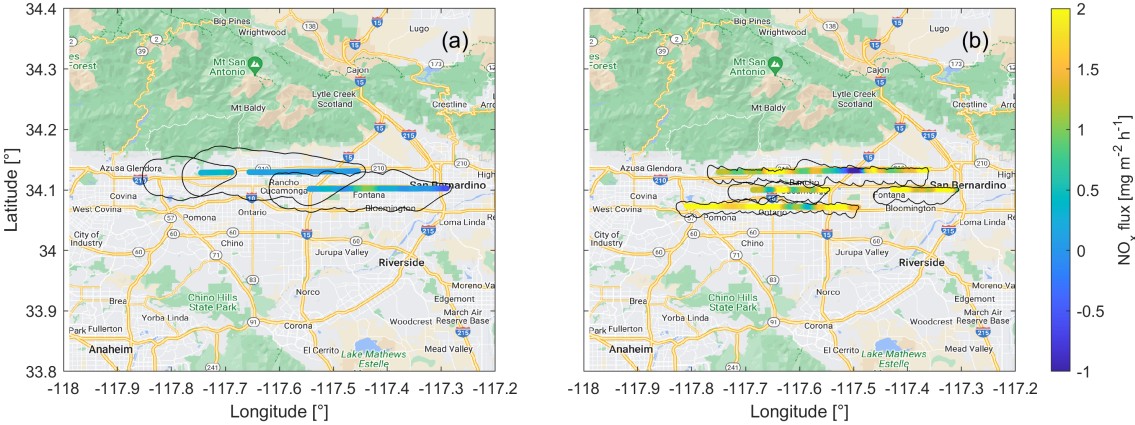

**Figure 4.** Flight segments in geographic proximity on two weekend days, (a) 6 June and (b) 12 June, with different footprint size. © Google Maps 2023.

footprint size for these segments was more than 4 times larger for 6 June with an average of $8.2 \pm 2.1$ km, than for 12 June with an average of $1.9 \pm 1.0$ km. The measured air on 12 June originated from above the highways in the San Bernardino valley. In

contrast, on 6 June we also captured air from adjacent, less $NO_x$ sources, such as residential areas, diluting the $NO_x$ polluted air from above the highways and leading to lower $NO_x$ fluxes. While all inputs for the footprint model were roughly similar (e.g. radar altitude 384 m (June 6) and 349 m (June 12); BLH 637 m (June 6) and 600 m (June 12)), the horizontal wind speed was significantly higher on 6 June with an average of $8.2 \pm 1.2$ m s$^{-1}$ than on June 12 where the average was $2.5 \pm 1.0$ m s$^{-1}$. This example also underlines the importance of footprint calculations in the interpretation of the observed fluxes.

## 2.7  Emission inventory

The California Air Resources Board (CARB) provides an emission inventory for Los Angeles with a 1 hour temporal resolution and a 4 km $\times$ 4 km spatial resolution over 12 vertical layers. The emission inventory is assembled from several sub-models accounting for different emission source categories. These are on-road vehicle emissions, aircraft emission and all other emissions, including for example shipping and port emissions. Mobile emissions are obtained via the ESTA (Emissions Spatial and

Temporal Allocator) model (CARB, 2019). Emissions from vehicles, including passenger cars, buses and heavy-duty trucks, are estimated via the EMFAC (EMission FACtor) model based on vehicle registrations and emission rate data for different vehicle types (CARB, 2021, 2022a). In combination with spatial information and temporal data, such as diurnal profiles and day-of-week dependence, the on-road emission inventory can be created via the ESTA model (CARB, 2019). The GATE (Gridded Aircraft Trajectory Emissions) model analogously provides spatially and temporally resolved aircraft emissions (CARB,

2017). Emissions from shipping and port activities, as well as additional point or area sources, are modeled with the SMOKE (Sparse Matrix Operator Kernel Emissions) model (CEMPD, 2022).





We are presenting a comparison of the $NO_x$ fluxes calculated from the RECAP-CA campaign measurements with the 2020 CARB emission inventory for Los Angeles, which is a baseline inventory and therefore does not include any effects related to the COVID-19 (coronavirus disease 2019) economic and social upheaval. For each flight day of the 2021 campaign, we include the according day of the week of the emission inventory in 2020. As 2020 was a leap year, the day of year of each considered flight is shifted by two from the 2020 calendar (e.g. Tuesday, 1 June, was the 152nd day of 2021 for which we consider Tuesday, 2 June, the 154th day of 2020, from the emission inventory). We combined the $500\,\text{m} \times 500\,\text{m}$ spatial resolution of the RECAP-CA $NO_x$ fluxes to the $4\,\text{km} \times 4\,\text{km}$ CARB grid for this comparison.

## 3  Results and Discussion

### 3.1  $NO_x$ emissions over Los Angeles

$NO_x$ concentrations and $NO_x$ fluxes over Los Angeles are separated into four different geographical regions, as shown in Figure 1b. We analyze the effects of temperature and the planetary boundary layer height, as well as differences between weekend and weekday data. Figure 5 shows the temperature for the different sections, measured on the research aircraft ($380 \pm 63\,\text{m}$ altitude). Lowest temperatures were observed in the coastal section with a median value of $17\,^{\circ}\text{C}$. Temperatures measured over Santa Ana and Downtown were slightly higher with median values around $19\,^{\circ}\text{C}$. Further inland, observed temperatures were highest with a median value of $24\,^{\circ}\text{C}$. Significant differences between the four sections were also observed regarding the boundary layer height (BLH) as presented in Figure 5b. The lowest BLH was found for the coastal section with a median value of $470\,\text{m}$, followed by Santa Ana with $490\,\text{m}$ and a median value of $540\,\text{m}$ for Downtown. The BLH in the San Bernardino valley was highest with a median value of $590\,\text{m}$. Figure 5c shows $NO_x$ concentrations and Figure 5d shows the corresponding fluxes over Los Angeles, separated into the geographical sections and into weekdays and weekends. Neither $NO_x$ concentrations nor $NO_x$ fluxes were found to be temperature dependent (see Figure S6).

Median concentrations were highest in Downtown with $0.011\,\text{mg}\,\text{m}^{-3}$ on weekdays and $0.006\,\text{mg}\,\text{m}^{-3}$ on weekends. In the San Bernardino valley, median concentrations were $0.010\,\text{mg}\,\text{m}^{-3}$ and $0.007\,\text{mg}\,\text{m}^{-3}$ on weekdays and weekends, respectively. The median measured fluxes in Downtown Los Angeles were $0.86$ and $0.35\,\text{mg}\,\text{m}^{-2}\,\text{h}^{-1}$, respectively for weekdays and weekends. In the San Bernardino valley the median fluxes were $0.97\,\text{mg}\,\text{m}^{-2}\,\text{h}^{-1}$ on weekdays and $0.46\,\text{mg}\,\text{m}^{-2}\,\text{h}^{-1}$ on weekends. In all of these locations weekend emissions decreased by 50 - 60 % from weekday values.

Concentrations were lower near the coast. Median $NO_x$ concentrations were similar for the coastal section and Santa Ana with around $0.005\,\text{mg}\,\text{m}^{-3}$ on weekdays. The weekend values were smaller with $0.003$ - $0.004\,\text{mg}\,\text{m}^{-3}$. However, no significant differences could be observed between weekday and weekend fluxes in these regions. Much of the coastal region is over water and the median values of fluxes are near zero on both weekdays and weekends and approximately $0.2\,\text{mg}\,\text{m}^{-2}\,\text{h}^{-1}$ on both weekdays and weekends in the Santa Ana region.

While $NO_x$ concentrations were observed to be highest over Downtown Los Angeles, $NO_x$ fluxes were found to be highest in the San Bernardino valley. This effect could be partly caused by the observed differences in the boundary layer height. While





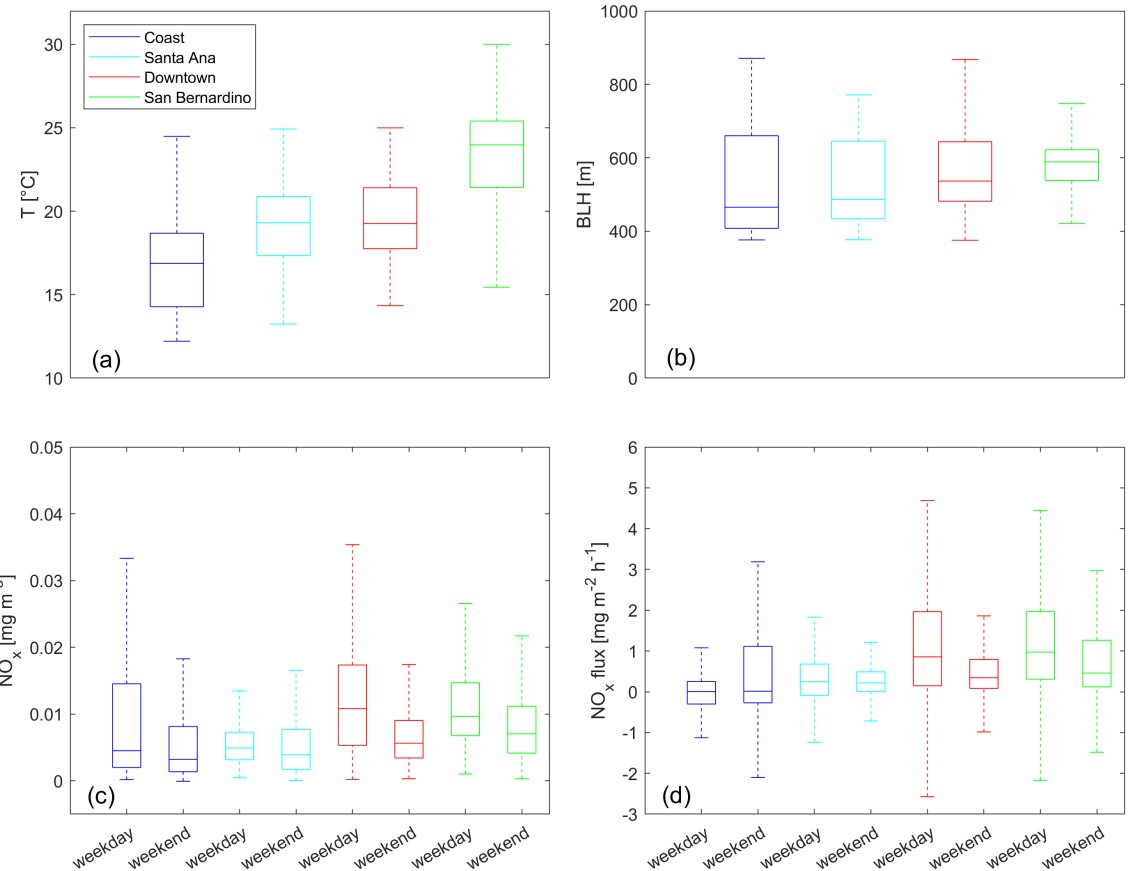

**Figure 5.** Boxplots for (a) the temperature, (b) the boundary layer height, (c) $NO_x$ concentrations and (d) $NO_x$ fluxes subdivided into four geographical sections according to Figure 1b and into weekday and weekend data. Please note that outliers are not shown.

highest emissions occurred in the San Bernardino valley, the increased planetary BLH (as shown in Figure 5b), should lead to
275 a $\sim 15\,\%$ lower mixing ratio. The differences in concentrations are likely also due to chemistry and advection.

Using the highway information by the California Department of Transportation (2015), we separated $NO_x$ fluxes into emissions from highway and non-highway grid cells. A $500\,\text{m} \times 500\,\text{m}$ grid cell is considered a highway emission when it is crossed by a highway. We present the resulting emission grid in Figure S7 of the Supplement. For weekdays, $NO_x$ fluxes from highway grid cells were on average $0.84 \pm 1.43\,\text{mg}\,\text{m}^{-2}\,\text{h}^{-1}$, approximately $25\,\%$ higher than $NO_x$ fluxes from
280 non-highway grid cells with an average of $0.67 \pm 1.26\,\text{mg}\,\text{m}^{-2}\,\text{h}^{-1}$. Weekend $NO_x$ fluxes from highways were on average $0.58 \pm 0.94\,\text{mg}\,\text{m}^{-2}\,\text{h}^{-1}$. Weekend $NO_x$ emissions from non-highway areas were $0.45 \pm 0.93\,\text{mg}\,\text{m}^{-2}\,\text{h}^{-1}$. Note the large $1\sigma$ standard deviations, indicating the large variability of the fluxes. The median values were lower compared to the mean values ($0.57$ and $0.29\,\text{mg}\,\text{m}^{-2}\,\text{h}^{-1}$ for highway and non-highway, respectively, on weekdays and $0.38$ and $0.26\,\text{mg}\,\text{m}^{-2}\,\text{h}^{-1}$ for highway and non-highway, respectively, on weekends), but showed a similar qualitative result with higher emissions from highway
285 compared to non-highway areas.





## 3.2 Comparison to the emission inventory

The comparison between the emission inventory and the calculated $NO_x$ fluxes is shown in Figure 6. We present the weekday data here and show the weekend data in Figure S8 of the Supplement. Panel (a) shows the RECAP-CA $NO_x$ fluxes at $4\,\mathrm{km} \times 4\,\mathrm{km}$, panel (b) presents the CARB emission inventory and in panel (c) we show the difference between the RECAP-CA and the CARB data according to Eq. (9).

$$\Delta NO_x flux = NO_x flux(RECAP) - NO_x flux(CARB) \tag{9}$$

As expected from the results presented in Section 3.1 the highest $NO_x$ fluxes were observed in the San Bernardino valley which is characterized by several heavily trafficked highways and warehouses that cause dense diesel truck traffic (Los Angeles Times, 2023). Elevated $NO_x$ emission also occurred in the region around Downtown Los Angeles. The average weekend RECAP-CA $NO_x$ fluxes (Figure S8a) showed a similar emission distribution over Los Angeles compared to the weekday data, but with smaller values. This is in line with the findings presented in Section 3.1.

Figure 6b shows average weekday $NO_x$ fluxes as predicted by the CARB emission inventory. The large $NO_x$ flux in proximity to the coast ($\sim 34.0\,°$ N, $118.4\,°$ W) with a value close to $8\,\mathrm{mg\,m^{-2}\,h^{-1}}$ was associated with aircraft emissions from Los Angeles International Airport (LAX). We show the $NO_x$ fluxes as predicted by CARB separated into (a) on-road emissions, (b) aircraft emissions, (c) area sources and (d) emissions from ocean going vessels in Figure S9 of the Supplement. Aircraft $NO_x$ emissions can also be observed in the San Bernardino valley ($\sim 34.1\,°$ N, $117.6\,°$ W) from Ontario International Airport which is illustrated in Figure S9b. High $NO_x$ fluxes in this area were also associated with on-road emissions, shown in Figure S9a. The Downtown Los Angeles area ($\sim 34.0\,°$ N, $118.2\,°$ W) also showed high fluxes which originated from on-road and area sources. Elevated $NO_x$ fluxes around Long Beach ($\sim 33.8\,°$ N, $118.2\,°$ W) were associated with shipping and port emissions. Average weekend $NO_x$ fluxes predicted by the emission inventory are presented in Figure S8b which showed a similar qualitative distribution compared to the weekday data, but were generally lower. Figure 6c presents the difference between the $NO_x$ fluxes from the RECAP-CA campaign and the CARB emission inventory. Red colors represent higher values for the RECAP-CA campaign compared to the emission inventory. Blue colors indicate higher fluxes from the emission inventory. In most places, the $NO_x$ fluxes predicted by the emission inventory were higher compared to the values from the RECAP-CA campaign. This difference was particularly pronounced in the area around Downtown Los Angeles and along the coast. We were not able to capture any airport emissions during the RECAP-CA campaign, as a result the differences in the vicinity of the Los Angeles and Ontario airports should not be interpreted as meaningful. $NO_x$ fluxes around Downtown Los Angeles are dominated by area sources and on-road emissions.

We observed higher $NO_x$ fluxes during the RECAP-CA campaign compared to the emission inventory in the San Bernardino valley. A possible explanation could be the accumulation of distribution and fulfillment centers which are accessible to delivery trucks via multiple highways in this area (Schorung and Lecourt, 2021; Los Angeles Times, 2023). Over the past two decades net sales via distribution centers have grown exponentially (Statista, 2022a). In the U.S., the number of delivered orders by the online retailer amazon has increased by nearly a factor of 6 between 2018 and 2020 (Statista, 2022b). $NO_x$ emissions in





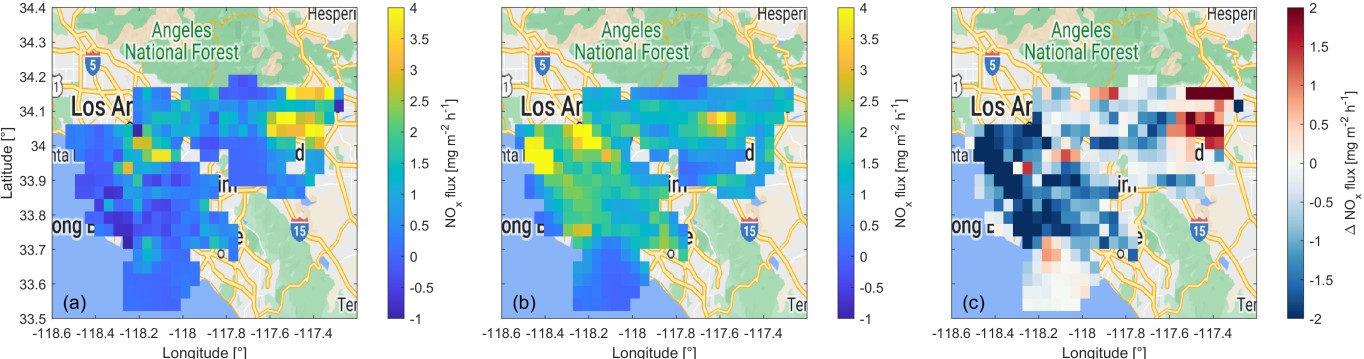

**Figure 6.** Weekday averages of NO$_x$ emissions across Los Angeles with a 4 km × 4 km spatial resolution (a) during the RECAP-CA campaign, (b) from the CARB emission inventory and (c) the difference between RECAP-CA and CARB NO$_x$ fluxes. © Google Maps 2023.

proximity to warehouses have likely increased to a similar extent in recent years which might not yet be incorporated in the
CARB 2020 emission inventory. Additional research is needed to examine more details of these differences and connect them
to specific processes in the inventory and observations.

In Figure S10 and S11 of the Supplement we show the NO$_x$ fluxes corrected for vertical divergence as presented in Section
2.5 in comparison to the CARB emission inventory for weekdays and weekends, respectively. The emission features shown in
Figure 6 for the RECAP-CA campaign are more pronounced after applying the factor for vertical correction. High emissions
are observed over Downtown Los Angeles and the inland highways in San Bernardino, while the coastal region and Santa Ana
show lower, and even negative fluxes. As a result, CARB emissions remain dominant over RECAP-CA fluxes in the coastal
region, but are lower around Downtown Los Angeles and in San Bernardino. The median values of the corrected fluxes are
around a factor of 3 higher compared to the non-corrected fluxes. The interquartile range increases by even more as a result
of the large scatter induced by the correction (compare Figure S5). This sensitivity analysis emphasizes how important the
characterization of the vertical flux divergence is and should be subject to future studies.

## 4   Conclusions

In this study, we have investigated NO$_x$ fluxes via wavelet analysis, based on in-situ observations of NO$_x$ concentrations and
the vertical wind speed during the research aircraft campaign RECAP-CA (Re-Evaluating the Chemistry of Air Pollutants in
CAlifornia) which took place in June 2021 over Los Angeles. We identified NO$_x$ concentrations to be highest over Downtown
Los Angeles, while we found highest NO$_x$ fluxes in the San Bernardino valley where a high planetary BLH induced a higher
dilution of the emitted NO$_x$. Both NO$_x$ concentrations and NO$_x$ fluxes revealed a weekend effect with higher values on week-
days due to more commuter traffic and more diesel trucks on roads, which was most pronounced over Downtown Los Angeles
and the San Bernardino valley. Footprint calculations revealed that the distance of the 90 % influence was on average 4 km,
whereby the horizontal wind speed played a dominant role in the footprint size. NO$_x$ emissions predicted by the California



Air Resources Board (CARB) 2020 were in the same order of magnitude, but on average higher compared to the RECAP-CA $NO_x$ fluxes. Spatially, the emission inventory particularly overestimated the fluxes in coastal proximity and over Downtown Los Angeles. In contrast, the emission inventory underestimated the $NO_x$ fluxes over the Eastern part of the San Bernardino valley.

*Data availability.* Data measured during the flight campaign, computed fluxes and footprints are available at https://doi.org/10.5281/zenodo.7786409
(Nussbaumer et al., 2023).

*Author contributions.* CMN analyzed the data with contributions from BKP, QZ and EYP. CMN wrote the manuscript. All authors contributed to designing the study and proofreading the manuscript.

*Competing interests.* At least one of the (co-)authors is a member of the editorial board of Atmospheric Chemistry and Physics.

*Acknowledgements.* We acknowledge Horst Fischer and Lenard Röder for valuable discussions and proofreading of this manuscript. The
RECAP-CA aircraft campaign was funded by the California Air Resources Board (20RD003 and 20AQP012) and the South Coast Air Quality Management District (#20327). The wavelet software was provided by C. Torrence and G. Compo, and is available at URL: http://paos.colorado.edu/research/wavelets/. Plots including Google Maps data were created with a matlab function by Bar-Yehuda, Z. (2022). This work was supported by the Max Planck Graduate Center with the Johannes Gutenberg-Universität Mainz (MPGC).



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
