# Peer review of "Figure S1. NPS Twin Otter aircraft, including the mounted inlet for air sampling."

_EGUsphere, 2023_

## Referee Comment (RC2)

**Review of "Measurement report: Airborne measurements of NOx fluxes over Los Angeles during the RECAP-CA 2021 campaign," Nussbaumer et al., ACP (2023)**

**Summary**

This paper presents airborne observations of NOx fluxes over Los Angeles during June 2021. Flux and footprint calculations are discussed in detail. Some analysis is presented regarding spatial and temporal variability, and a comparison against an emission inventory shows under and over-prediction at different locations. The writing is generally clear, and the number and style of figures is appropriate. The degree of analysis is sufficient for a measurement report.

Publication is recommended after consideration of the following comments, which I would characterize as "minor" because there are no fatal flaws.

**General Comments**

Lag-covariance, cospectra, and detrending: In two of the three cases in Figs S2/S3, it is difficult to identify a clear lag peak even for temperature. The cospectra do not look like canonical boundary layer turbulence (see Kaimal 1972/76) with multiple peaks at low frequencies. I have a few suggestions on how to deal with this. First, was the NOx data detrended prior to calculating these? While this is not strictly necessary for wavelet fluxes, I have found it helps to remove non-turbulent low-frequency variability when generating these plots, especially the lag covariance. Second, in Sect. 2 (possibly in a new subsection) it would be prudent to add some explicit discussion of these plots and the implications for data quality and limitations – especially since this is a "Measurement Report." Also, for airborne fluxes it is more appropriate to use a length scale for cospectra (L = aircraft speed / frequency).

NOx units: mg m$^{-3}$ is a non-standard unit for atmospheric chemistry. Please use mixing ratio units (e.g., ppbv).

Archived Data: For anyone else who wants to use this data, this needs a little work.

- A text file is fine, but ideally this would be in ICARTT format (https://www.earthdata.nasa.gov/esdis/esco/standards-and-practices/icartt-file-format) or similar for easy sharing and also to ensure appropriate metadata is included. Some of that information is in the manuscript, but it should be in the data file too.
- Would be wise to include a link or DOI for the paper in the data file.
- Pressure or GPS altitude should be included in addition to radar altitude
- The raw NOx flux should be included as well as the moving average.
- Temperature fluxes?
- The "footprints" file appears to just contain an index to footprints. In this case, I feel like it could just be included in the main data file? Also, I know the footprints are likely to large (data-wise) to archive, but it would help to provide some information to users on how they can get them (even if it

means contacting you). Alternatively, you can provide sufficient data in the file for people to calculate footprints themselves.

**Specific Comments**

L9: quantify what is meant by "too high" and "too low."

L67: by sampling speed, do you mean aircraft speed, or the speed of air in the inlet? Aircraft speed is more relevant for effective spatial resolution.

L87: Is a 10-second average detection limit relevant, given typical turbulence scales?

L132: How do you determine the boundary layer depth? How uniform is it spatially (or how uniform do you assume it is)?

L180: What are uncertainties in m and c? Are uncertainties in the divergence fit propagated to fluxes?

L187: What is the reason for the 20% choice? Seems arbitrary as written.

L189: "where the measured air masses originated" is not quite right. It is more like the area over which surface source and sinks influence the observed flux. Suggest rephrasing.

L266: is the weekday/weekend difference consistent with expectations from prior observations and literature (e.g., concentrations at ground or airborne)?

L271: Could you have sub-selected the fluxes for footprints over land only? Not suggesting you do this now, but a recommendation for future work.

L291: this equation seems unnecessary. Might be easier to change the colorbar label in Fig. 6c to say RECAP – CARB? Also, would it make more sense to reverse this so that inventory overprediction is red?

L322: This threw me off, as I had assumed that Fig. 6 was utilizing divergence-corrected fluxes. Suggest clarifying this at the beginning of Sect 3.2.

Conclusions: this section is short. I realize this is a measurement report, but this is an opportunity to provide guidance for future work. What else could be done with this data? What data are you missing that you would want if we did this experiment again? How would you do it differently?

**Technical Comments**

L12: (NO2),

L27: provides (the verb is referencing a single item "the combination")

L28: the phrase starting with "that describe" is oddly-worded, and this first sentence is very long. Suggest breaking up and rewording.

L56: identified as a key

L67: approximately

L74: nitpicky point, but the coastal legs seems to be NW – SE.

L100: using omega for vertical wind is atypical, I think. Small w is more standard.

L106: replace "steady state conditions" with "stationarity." These are not synonymous, especially in the context of chemistry. Also on L109.

L106: similarly, but "homogeneous horizontal air masses" do you mean ergodicity?

L123: integrated across scales

L125: "Eq. (6) where . . .". Also, it might be easier for readers to put variable descriptions after the equation.

L165: suggest deleting "mostly through differing wind speeds with altitude."

L170: "in dependence of" is not the right phrasing. Maybe something like "flux vertical profile in a BLH-normalized coordinate system"? But that feels awkward too.

L207: define $z_0$.

Figure 4: consider adding arrows for mean wind direction.

L256: delete "significant." Let readers decide.

Figure 5: define components of box plots in caption.

L292: Section 3.1,

L318: Amazon (capitalize)

L333: no need to redefine RECAP

L334: downtown (lower case)

L338: 4 km upwind

---

## Author Comment (AC1)

**Referee 1**

Clara M. Nussbaumer et al. presented the NOx fluxes from airborne measurements in Los Angeles during the aircraft campaign RECAP-CA. They showed both NOx concentrations and fluxes were higher in the weekdays and lower in the weekend. They also showed the difference between their calculated NOx fluxed and NOx emissions from the CARB inventory. The observations are valuable and very useful to the emission community. The paper is generally well rewritten. However, I still have some minor concerns before it can be published.

We would like to thank Referee 1 for taking the time to review our manuscript and the valuable feedback. We have corrected our manuscript according to the referee's comments and think it is now improved.

line 94: please add what is NOy, what species are included in the NOy?

$NO_y$ describes the sum of all reactive nitrogen species including $NO_x$ and higher nitrogen oxides: $HNO_3$, HONO, peroxy nitrates ($RO_2NO_2$), alkly nitrates ($RONO_2$), etc. We have clarified this in the manuscript.

Lines 98 ff.: Reactive nitrogen species ($NO_y \equiv NO_x$, $HNO_3$, HONO, $RONO_2$, $RO_2NO_2$, ...) were detected through thermal dissociation at ~500°C to $NO_2$ in the third channel (Day et al., 2002).

line 160-162: it was mentioned that the values of the NOx flux are dominated by the atmospheric variability. Can you explain a little more about it?

The uncertainty of the nitrogen oxides measurements used for calculating the NOx fluxes over Los Angeles is not dominated by the measurement uncertainty (typically around 7% for the used instrument), but rather by the atmospheric variability (around 30% for this study) induced by factors like varying meteorological conditions and the time-of-day. Additional uncertainties arise from the method applied (wavelet transformation) for determining the NOx emissions, for which a detailed error analysis is provided in Zhu et al. (2023). We agree that this was worded somewhat confusingly in the text and have rephrased it for clarification.

Lines 173 ff.: The overall uncertainty of the calculated $NO_x$ flux is composed of the uncertainty of the measurement of the $NO_x$ concentration and the vertical wind speed. We find that the $NO_x$ median and average values are dominated by the atmospheric variability and not the measurement uncertainties. The observed atmospheric variability of $NO_x$ is in the order of 30% ($1\sigma$) which is around 4 times higher than the instrumental precision of <7% ($1\sigma$). Additional uncertainty is associated with the presented method of performing the wavelet transformation, including random and systematic errors (Lenschow et al., 1994; Mann and Lenschow, 1994; Wolfe et al., 2015; Vaughan et al., 2021). A detailed error analysis for these observations is provided in Zhu et al. (2023).

line 174 you are using the boundary layer height, where did you get the boundary layer height? Is it measured or modeled boundary layer height? What is the uncertainty of the boundary layer height?

The boundary layer height was determined from changes in water vapor, the dew point and toluene concentrations, which are usually high in the boundary layer and decrease promptly in the free troposphere. We have clarified this in the text.

Line 85 ff.: The planetary boundary layer (PBL) height was determined from changes in water vapor and toluene concentrations, the dew point and temperature, which decrease rapidly at the boundary between the BL and the free troposphere (Pfannerstill et al., 2023). The aircraft crossed the top of the PBL at several times during each flight providing these direct observations.

Line 177 'the fit' is the the linear fit of Fz and z/zi. Please mention it here.

We have added this information in the text.

Line 193 ff.: In order to investigate the influence of vertical divergence, we compare an analysis with a correction of the fluxes using the linear fit of the $NO_x$ flux ($F_z$) and the dimensionless altitude ($z/z_i$) as shown in Figure S4 to our analysis assuming the divergence is zero, which we will refer to as 'sensitivity study' in the following.

Line 187. Please make it more clear what is the sensitivity study. Can you also provide a figure of the vertical divergence versus the dimensionless which excluding data points within the upper 20 % of the boundary layer? I get very confused by looking at Figure s4 and Figure s5. It would be nice if you also use different colors to indicate the density of the points.

With the sensitivity study, we attempt to investigate the impact of a range of choices for vertical flux divergence (including zero) on the interpretation of our measurements. With the available data, unfortunately we cannot perform an unambiguous correction of the flux divergence, whose existence is indicated by the Figure S4 (plotting the flux versus the dimensionless altitude). However, we apply a correction factor derived from the linear fit of $F_z$ versus $z/z_i$ to show the potential effect of the vertical divergence compared to analyses which assume the divergence to be zero. We have clarified what we mean with 'sensitivity study' in the text.

Lines 193 ff.: In order to investigate the influence of vertical divergence, we compare an analysis with a correction of the fluxes using the linear fit of the $NO_x$ flux ($F_z$) and the dimensionless altitude ($z/z_i$) as shown in Figure S4 to our analysis assuming the divergence is zero, which we will refer to as 'sensitivity study' in the following.

We have added Figure of $F_z$ vs $z/z_i$, excluding data point in the upper 20% of the boundary layer to Figure S 7 of the Supplement. The resulting linear fit shows a more vertical course (to be expected after vertical divergence correction) compared to Figure S6 which included data points throughout the entire boundary layer.

[Figure]

**Figure S7.** Dimensionless altitude $z/z_i$ versus the corrected $NO_x$ flux according to Figure S4, omitting data in the upper 20% of the boundary layer, which are most strongly affected by uncertainties in the vertical divergence correction. Black dots represent all data points. The green dashed line shows the linear fit of all data points. The red points and error bars represent the binned means with the $1\sigma$ variability.

Figure S4 presents all available data points of the calculated $NO_x$ flux versus the dimensionless altitude. We take the resulting linear fit to correct the NOx fluxes for vertical divergence and the result is presented in Figure S6 (S5 before). Due to large uncertainties of the correction in the upper 20% of the BL we omit these data in the following sensitivity study. We now show the corrected fluxes for the lower 80% of the BL in Figure S7 of the Supplement. We have also created a density plot to indicate the data distribution which we show in Figure S5.

[Figure]

**Figure S5.** Density plot of the dimensionless altitude $z/z_i$ versus the $NO_x$ flux to show the distribution of the data presented in Figure S4.

Line 195-199: the footprint calculation is dependent on many variables and using the KL04 model. What are the meteo input for the model? Do you use measured data or data from meteo models?

The meteorological inputs for the footprint calculations including the wind direction, the crosswind fluctuations and the vertical wind fluctuations were obtained via a radome flow angle probe which provided 3D wind data. The aircraft's altitude was measured via a C-MIGITS. These measurements are described in detail in Karl et al. 2013). The boundary layer height was determined as described above via changes of water vapor, toluene, temperature and dew point. We have added a reference to Section 2.2, where we describe the acquisition of the meteorological data.

Lines 222 f.: Please find details on the acquisition of the meteorological inputs in Section 2.2.

Figure 4. What do the black lines indicate in figure 4? Are they the flight paths? Please mention it in the caption.

The black lines indicate the contour of the 90% footprints. The flight paths are colored by the calculated $NO_x$ flux. We have clarified this in the Figure caption and in the text.

**Figure 4.** Flight segments colored by the $NO_x$ flux in geographic proximity on two weekend days, (a) 6 June and (b) 12 June, with different footprint size. The black lines represent the contour of the 90 % footprints. © Google Maps 2023.

Lines 245 f.: The two panels present selected flight segments colored by the NOx flux in geographic proximity (…)

Lines 249 f.: At the same time, the footprint size for these segments, represented by the black lines, was more than 4 times larger (…)

Line 300, what are 'area emissions'?

Area emissions are fluxes that originate from a larger area, instead of a point source e.g. an industrial facility. They usually represent small individual emissions which accumulate to have a significant contribution. For $NO_x$, area sources mostly include residential fuel combustion processes such as heating or cooking. We have indicated this in the manuscript and added a reference by the California Air Resources Board.

Lines 331 ff.: We show the $NO_x$ fluxes as predicted by CARB separated into (a) on-road emissions, (b) aircraft emissions, (c) area sources (e.g. residential heating or cooking emissions which accumulate over a larger area (CARB, 2023)) and (d) emissions from ocean going vessels in Figure S11 of the Supplement.

Line 310-313, I don't understand this part. The airport emissions were not captured by the measurements. How about comparing your NOx fluxes with the CARB emissions excluding aircraft emissions?

The airport emissions were likely only captured to a small extent, which can be seen when looking at the footprints, e.g. around LAX:

[Figure]

The CARB aircraft emission inventory also includes aircraft emissions in further distance to the airport, as well as ground handling equipment and vehicle traffic around the airport. We agree that this was not clear in the text and we have clarified this.

Lines 328 ff.: The large NOx flux in proximity to the coast (~34.0°N, 118.4°W) with a value close to 3.5 mg m$^{-2}$ h$^{-1}$ was associated with aircraft emissions, as well as ground handling equipment and vehicle traffic, from and around Los Angeles International Airport (LAX). Additionally, emissions from aircraft not only at the surface but also at elevated altitudes could contribute to the observed value.

and

Lines 344 ff.: Due to lively air traffic, the research aircraft could not approach the airport closely and the footprints only covered a minor area of LAX airport. As a

result, the differences in the vicinity of the airport should not be interpreted as meaningful.

Line 325-330: In Figure 6 and Figure s10, NOx fluxes are quite different. Which one shows better results? Is it necessary to include the correction of vertical divergence in the flux calculation? Why are the emissions enhanced over Downtown Los Angeles and the inland highways in San Bernardino, but lower in the coastal region and Santa Ana? Please Add more discussion about the influence of vertical divergence.

Figure S12 (previously Figure S10) represents the results of a sensitivity study in an attempt to investigate the influence of vertical flux divergence and underlines that this effect could be quite large. While Figure S4 ($F_z$ vs $z/z_i$) indicate that vertical flux divergence can play a role for example through entrainment from above or horizontal advection, we do not perform a correction of the calculated $NO_x$ fluxes because the correlation between the flux and the dimensionless altitude does not provide significant results. The linear fit of $F_z$ vs $z/z_i$ exhibits an $R^2$ of only 6% which likely arises from the surface heterogeneity experienced over Los Angeles. This includes heterogeneity in time (e.g. rush-hour traffic) and space (a high variety of sources). Therefore, the results shown in Figure S12 should not be interpreted as unambiguously supporting a specific value for the flux divergence, rather only an idea of the impact of vertical divergence. Unfortunately, with the available data set, we cannot convincingly determine the flux divergence over Los Angeles, and we have to acknowledge this drawback in our analysis. We therefore strongly suggest the characterization of vertical flux divergence over heterogeneous sources to be subject to future studies. We have added some discussion in the manuscript regarding this topic.

Lines 363 ff.: We do not correct the fluxes for vertical divergence as our data set does not provide significant or unambiguous indication for its occurrence and extent. This is likely an outcome of the source heterogeneity experienced across Los Angeles as most emissions are highly variable in time and space. In previous studies, the vertical divergence has been successfully characterized via the correlation of the flux and the dimensionless altitude over homogeneous surfaces, which is not applicable to Los Angeles. Instead, carefully planned stacked race track flights could provide insights into vertical flux divergence. This sensitivity analysis emphasizes how important the characterization of the vertical flux divergence is and should be subject to future studies.

Line 332 change 'in-situ' to 'airborne'

We have changed this.

Section 4: The conclusion section is only a short summary of the results. Please also indicate the implication of the study. What can we learn from the difference between the estimated NOx fluxes and the CARB inventory? What is your conclusion after investigating the influence of vertical divergence. Also discuss the limitation of the study and recommendations for future study.

We have added some discussions in the conclusion section.

Lines 379 ff.: Spatially, the emission inventory overestimated the fluxes in coastal proximity and over Downtown Los Angeles, which could be due to COVID-19 related reductions, such as a shift to more remote work and less commuter traffic, general emission reductions not yet captured by the emission inventory, or misallocation of emission sources in the inventory. In contrast, the emission inventory underestimated the $NO_x$ fluxes over the Eastern part of the San Bernardino valley where an increased activity of trucks going to and from warehouses due to the exponential growth of online retailers, such as Amazon, lead to higher $NO_x$ emissions in recent years. A single uniform correction for vertical divergence could locally lead to improved agreement in this part of the domain, but would at the same time increase the difference in other parts of the studied area. Being an important tool in air quality regulation, we encourage further investigation of the accuracy of local emission inventories with observations from aircraft, towers or dense networks. For flux measurements from aircraft or towers, a particular focus on improving vertical divergence characterization, in order to provide accurate emission predictions would be especially beneficial.

---

## Author Comment (AC2)

**Referee 2**

Review of "Measurement report: Airborne measurements of NOx fluxes over Los Angeles during the RECAP-CA 2021 campaign," Nussbaumer et al., ACP (2023)

Summary

This paper presents airborne observations of NOx fluxes over Los Angeles during June 2021. Flux and footprint calculations are discussed in detail. Some analysis is presented regarding spatial and temporal variability, and a comparison against an emission inventory shows under and over-prediction at different locations. The writing is generally clear, and the number and style of figures is appropriate. The degree of analysis is sufficient for a measurement report. Publication is recommended after consideration of the following comments, which I would characterize as "minor" because there are no fatal flaws.

We would like to thank Glenn Wolfe (Referee 2) for taking the time to review our manuscript and the positive feedback. We have corrected the manuscript according to his comments.

General Comments

Lag-covariance, cospectra, and detrending: In two of the three cases in Figs S2/S3, it is difficult to identify a clear lag peak even for temperature. The cospectra do not look like canonical boundary layer turbulence (see Kaimal 1972/76) with multiple peaks at low frequencies. I have a few suggestions on how to deal with this. First, was the NOx data detrended prior to calculating these? While this is not strictly necessary for wavelet fluxes, I have found it helps to remove non-turbulent low-frequency variability when generating these plots, especially the lag covariance. Second, in Sect. 2 (possibly in a new subsection) it would be prudent to add some explicit discussion of these plots and the implications for data quality and limitations – especially since this is a "Measurement Report." Also, for airborne fluxes it is more appropriate to use a length scale for cospectra (L = aircraft speed / frequency).

We have explored the data detrending, but found that it did not help much. Instead, we have added some discussion of the lag-covariance and co-spectra plots. We have also added the length scale to the co-spectra.

[Figure]

Lines 159 ff.: In Figure S2 of the Supplement we present an example covariance peak for NO$_x$ and potential temperature θ with the vertical wind speed, respectively, for three segments on June 6. For all three segments shown here the covariance for NO$_x$ and vertical windspeed is clearly identifiable. The covariance peak for θ and the vertical windspeed can only be determined for two of these segments (middle and right panel). However, particularly for NO$_x$, the identified lag times match quite well. As we expect the lag time not to vary throughout one flight (its variation is primarily associated with alignment of different computer clocks and not variation in the transit time to the detection point), we correct all segments with the median lag time of the identified segments. We show an example co-spectrum for the NO$_x$ flux and the heat flux in Figure S3 of the Supplement, for three segments on June 6 corresponding to Figure S2. The Nyquist frequency which is equal to half the sampling frequency is shown as black dotted lines. We were able to capture most eddies due to the high sampling frequency of 5Hz. Similar to the lag-covariance however, we observe difficulties for some of the segments. We explicitly chose positive and negative examples of lag-covariance and co-spectra plots here to underline the strength, but also the limitation of our data quality which varies from segment to segment and is dependent on various factors, including the instrumental performance, meteorological conditions, the relative aircraft position within the boundary layer, the aircraft speed, the segment length, changes in altitude and the roll angle of the aircraft. A detailed error analysis and discussion can be found in Zhu et al. (2023).

NOx units: mg m-3 is a non-standard unit for atmospheric chemistry. Please use mixing ratio units (e.g., ppbv).

We have updated Figure 2a) and 4c) from mg m$^{-3}$ to ppbv and have changed the details in the text.

Figure 2a):

[Figure]

Figure 4c):

[Figure]

Lines 289 ff.: Median mixing ratios were highest in Downtown with 6.5ppbv on weekdays and 3.4ppbv on weekends. In the San Bernardino valley, median concentrations were 5.8ppbv and 4.1ppbv on weekdays and weekends, respectively.

and

Lines 296 ff.: Mixing ratios were lower near the coast. Median $NO_x$ mixing ratios were similar for the coastal section and Santa Ana with 2.8-3.0ppbv on weekdays. The weekend values were smaller with 2.0-2.3ppbv.

We have also changed all flux units to mg N $m^{-2}$ $h^{-1}$, as we found that the $NO_x$ flux in mg $m^{-2}$ $h^{-1}$ is dependent on the ratio of NO to $NO_2$ which arises from the unit conversion from ppbv to mg $m^{-3}$.

Archived Data: For anyone else who wants to use this data, this needs a little work.

- A text file is fine, but ideally this would be in ICARTT format (https://www.earthdata.nasa.gov/esdis/esco/standards-and-practices/icartt-file-format) or similar for easy sharing and also to ensure appropriate metadata is included. Some of that information is in the manuscript, but it should be in the data file too.
- Would be wise to include a link or DOI for the paper in the data file.
- Pressure or GPS altitude should be included in addition to radar altitude
- The raw NOx flux should be included as well as the moving average.
- Temperature fluxes?

- The "footprints" file appears to just contain an index to footprints. In this case, I feel like it could just be included in the main data file? Also, I know the footprints are likely to large (data-wise) to archive, but it would help to provide some information to users on how they can get them (even if it means contacting you). Alternatively, you can provide sufficient data in the file for people to calculate footprints themselves.

Thanks for the suggestions. We have updated the data file according to these recommendations and uploaded a new version in ICARTT format: https://doi.org/10.5281/zenodo.8199013. Please note that we have still generated two separate files for each day as the footprint shapes cannot be resolved on the same time series as the remaining data. The first file contains $NO_x$ mixing ratios, $NO_x$ fluxes and the relevant meteorological data and the second file contains the shapes of the footprints which were generated via the KL04+2D algorithm. We have added a note that readers can request other file formats, e.g. different resolutions, through contacting the corresponding authors. We have added the DOI for the paper in the title of the Supporting data, which can be edited once the paper is published and receives a new DOI.

Line 391: Additional files, e.g. in different resolutions, are available from the corresponding authors upon request.

Specific Comments

L9: quantify what is meant by "too high" and "too low."

We have changed this wording to "higher than expected" and "lower than expected".

Lines 8 ff.: The comparison of the RECAP-CA and the modeled CARB NOx fluxes suggest the modeled emissions are higher than expected near the coast and in downtown Los Angeles and lower than expected further inland in the Eastern part of the San Bernardino valley.

L67: by sampling speed, do you mean aircraft speed, or the speed of air in the inlet? Aircraft speed is more relevant for effective spatial resolution.

We are referring to the aircraft speed. We have clarified this in the text.

Lines 66 f.: The ambient air was sampled with an inlet approx. 1 m above the aircraft nose at a sampling speed (aircraft speed) of around 60 m s$^{-1}$.

L87: Is a 10-second average detection limit relevant, given typical turbulence scales?

This is correct. We have added an information that the detection limit of the instrument is higher for higher resolved data.

L132: How do you determine the boundary layer depth? How uniform is it spatially (or how uniform do you assume it is)?

The boundary layer height was determined from changes in water vapor, the dew point and toluene concentrations, which are usually high in the boundary layer and decrease promptly in the free troposphere. We have clarified this in the text.

Line 85 ff.: The planetary boundary layer (PBL) height was determined from changes in water vapor and toluene concentrations, the dew point and temperature, which

decrease rapidly at the boundary between the BL and the free troposphere (Pfannerstill et al., 2023). The aircraft crossed the top of the PBL at several times during each flight providing these direct observations.

L180: What are uncertainties in m and c? Are uncertainties in the divergence fit propagated to fluxes?

The linear fit of the flux versus the dimensionless altitude exhibits an $R^2$ of only 6% and therefore does not provide significant evidence for the occurrence and extent of a vertical flux divergence. In turn, the overall uncertainty is dominated by the uncertainty that arises from describing $F_z$ vs $z/z_i$ with a linear function, rather than from the uncertainties in slope and intercept. The fluxes shown in Figure S10, corrected with the information provided by this linear fit, should not be interpreted as meaningful and are only supposed to provide an idea of the impact of vertical divergence. We have added some text in the manuscript.

Lines 197 f.: The uncertainty of this correction is dominated by the uncertainty in the fitted line, which arises from describing $F_z$ vs $z/z_i$ with a linear function as presented in Figure S4.

L187: What is the reason for the 20% choice? Seems arbitrary as written.

The uncertainty of the correction of the vertical flux divergence strongly increases with decreasing distance to the boundary layer height for both mathematical and meteorological reasons (see Lines 187 ff.) and the correction produces unreasonable results (multiple values up to -50 and +50 mg m$^{-2}$ h$^{-1}$ and individual values significantly higher). We therefore decided to omit data close to the BLH and the 20% value is a compromise between increasing uncertainty with altitude and the resulting data loss. An additional reason is that the upper 20% of the BL is likely to be influenced by entrainment (Druilhet and Durand (1984) and Stull (1988)). As this choice is still an estimate, we would like to emphasize that we only perform the flux correction and the omission of data points within the upper 20% of the boundary layer as part of a sensitivity study to investigate the impact of vertical divergence. We have clarified this in the manuscript.

Lines 206 ff.: Thus, for the sensitivity study, we omit data points within the upper 20 % of the boundary layer ($z/z_i \geq 0.8$), as a trade-off between high uncertainties close to the top of the PBL and the associated data loss. Additionally, the upper 20% of the boundary layer were found to be most likely influenced by entrainment (Druilhet and Durand, 1984; Stull, 1988).

L189: "where the measured air masses originated" is not quite right. It is more like the area over which surface source and sinks influence the observed flux. Suggest rephrasing.

Thanks for pointing this out. We have rephrased this statement.

Lines 212 f.: In order to map emissions, we performed footprint calculations which help to identify the areas over which the associated sources and sinks influence the observed fluxes (Vesala et al., 2008).

L266: is the weekday/weekend difference consistent with expectations from prior observations and literature (e.g., concentrations at ground or airborne)?

In Nussbaumer and Cohen (2020), we have previously analyzed the weekday/weekend changes in $NO_x$ mixing ratios in Los Angeles based on ground-based observations and found a reduction between 30 and 40% from weekdays to weekends which is in line with our results for Downtown and the San Bernardino valley. We have added the reference and a note in the manuscript.

Lines 290 f.: The observed levels of $NO_x$ reductions from weekdays to weekends are consistent with previous results based on ground-based measurements across Los Angeles, which, for example, we have investigated in Nussbaumer and Cohen (2020).

L271: Could you have sub-selected the fluxes for footprints over land only? Not suggesting you do this now, but a recommendation for future work.

Thank you for this suggestion. We were hoping to capture some shipping emissions, but the number of footprints over water in our study might be too small and too broad to capture the line sources that ships represent. We will keep this in mind for future studies.

L291: this equation seems unnecessary. Might be easier to change the colorbar label in Fig. 6c to say RECAP – CARB? Also, would it make more sense to reverse this so that inventory overprediction is red?

We have updated Figure 6 (and Figure S10) and the text according to these suggestions.

[Figure]

Lines 320 ff.: (…) and in panel (c) we show the difference between the CARB and the RECAP-CA data. Red colors indicate higher fluxes from the emission inventory and blue colors show higher fluxes from the RECAP-CA airborne measurements.

L322: This threw me off, as I had assumed that Fig. 6 was utilizing divergence-corrected fluxes. Suggest clarifying this at the beginning of Sect 3.2.

Thanks for pointing this out. We have added some text for clarification at the beginning of Section 3.2 and a more detailed explanation for this choice when discussing the differences in corrected and non-corrected fluxes at the end of Section 3.2.

Lines 317 ff.: This Figure presents measured $NO_x$ fluxes, which are not corrected for vertical divergence. We present the results of the sensitivity study (as described in Section (2.5)) applying a correction according to the linear fit presented in Figure S4 and discuss the implications of this correction at the end of this Section.

and

Lines 363 ff.: We do not correct the fluxes for vertical divergence as our data set does not provide significant or unambiguous indication for its occurrence and extent. This is likely an outcome of the source heterogeneity experienced across Los Angeles as most emissions are highly variable in time and space. In previous studies, the vertical divergence has been successfully characterized via the correlation of the flux and the dimensionless altitude over homogeneous surfaces, which is not applicable to Los Angeles. Instead, carefully planned stacked race track flights could provide insights into vertical flux divergence. This sensitivity analysis emphasizes how important the characterization of the vertical flux divergence is and should be subject to future studies.

Conclusions: this section is short. I realize this is a measurement report, but this is an opportunity to provide guidance for future work. What else could be done with this data? What data are you missing that you would want if we did this experiment again? How would you do it differently?

We have added some discussions in the conclusion section.

Lines 379 ff.: Spatially, the emission inventory overestimated the fluxes in coastal proximity and over Downtown Los Angeles, which could be due to COVID-19 related reductions, such as a shift to more remote work and less commuter traffic, general emission reductions not yet captured by the emission inventory, or misallocation of emission sources in the inventory. In contrast, the emission inventory underestimated the $NO_x$ fluxes over the Eastern part of the San Bernardino valley where an increased activity of trucks going to and from warehouses due to the exponential growth of online retailers, such as Amazon, lead to higher $NO_x$ emissions in recent years. A single uniform correction for vertical divergence could locally lead to improved agreement in this part of the domain, but would at the same time increase the difference in other parts of the studied area. Being an important tool in air quality regulation, we encourage further investigation of the accuracy of local emission inventories with observations from aircraft, towers or dense networks. For flux measurements from aircraft or towers, a particular focus on improving vertical divergence characterization, in order to provide accurate emission predictions would be especially beneficial.

Technical Comments

L12: (NO2),

We have corrected this.

L27: provides (the verb is referencing a single item "the combination")

Thanks, corrected.

L28: the phrase starting with "that describe" is oddly-worded, and this first sentence is very long. Suggest breaking up and rewording.

We have reworded the sentence.

Lines 27 ff.: The combination of emission inventories with models provides insight into the emission reductions needed to achieve healthy air. It helps us understand

atmospheric dynamics which includes the transport of emitted trace gases from their source through the atmosphere, their deposition to the earth's surface and the oxidation processes they are involved in.

L56: identified as a key

Corrected.

L67: approximately

Changed.

L74: nitpicky point, but the coastal legs seems to be NW – SE.

Corrected to 'Northwest-Southeast legs'

L100: using omega for vertical wind is atypical, I think. Small w is more standard.

Changed all $\omega$ to $w$

L106: replace "steady state conditions" with "stationarity." These are not synonymous, especially in the context of chemistry. Also on L109.

Changed steady state to stationary for both cases.

L106: similarly, but "homogeneous horizontal air masses" do you mean ergodicity?

We have replaced this term by "vertically homogeneously mixed boundary layer".

Lines 111 f.: Requirements for accurate fluxes with EC are stationary conditions and a vertically homogeneously mixed boundary layer.

L123: integrated across scales

Changed.

L125: "Eq. (6) where . . .". Also, it might be easier for readers to put variable descriptions after the equation.

We agree and have moved the variable descriptions behind the equation.

L165: suggest deleting "mostly through differing wind speeds with altitude."

Deleted.

L170: "in dependence of" is not the right phrasing. Maybe something like "flux vertical profile in a BLH-normalized coordinate system"? But that feels awkward too.

We have rephrased this to:

Lines 186 ff.: A different approach is described in Wolfe et al. (2018) and Zhu et al. (2023) which investigates the vertical profile of the calculated $NO_x$ fluxes, normalized to the boundary layer height over a homogeneous surface.

L207: define z0.

Added.

Figure 4: consider adding arrows for mean wind direction.

The wind direction is very similar for both days with a median of 266° on June 6 and a median of 250° on June 12. This can also be seen when looking at the frequency

distribution of the wind direction on these days. The distribution for June 12 is broader (also more data points), but the histograms peak at similar values for the wind direction. As the wind direction does not seem to be a deciding factor, we would prefer to not show the arrows in the figure for better clarity.

[Figure]

L256: delete "significant." Let readers decide.

Deleted.

Figure 5: define components of box plots in caption.

We have added a description of the colors in the caption of the Figure.

L292: Section 3.1,

Corrected.

L318: Amazon (capitalize)

Changed.

L333: no need to redefine RECAP

Deleted.

L334: downtown (lower case)

We have capitalized Downtown throughout the entire manuscript as it describes a custom-made area according to Figure 1b. Please correct, if wrong.

L338: 4 km upwind

Added.